# Differential angiogenesis of bone and muscle endothelium in aging and inflammatory processes

Chiara Arrigoni [1,2,3✉], Paola Ostano[4], Simone Bersini[1,2,3], Martina Crippa[1,2,5], Maria Vittoria Colombo[1,2,5], Mara Gilardi[6], Luigi Zagra[7], Maurizia Mello-Grand [4], Ilaria Gregnanin[4], Carmen Ghilardi[8], Maria Rosa Bani[8], Christian Candrian[2,3], Giovanna Chiorino [4] & Matteo Moretti[1,2,3,7]

Different tissues have different endothelial features, however, the implications of this heterogeneity in pathological responses are not clear yet. "Inflamm-aging" has been hypothesized as a possible trigger of diseases, including osteoarthritis (OA) and sarcopenia, often present in the same patient. To highlight a possible contribution of organ-specific endothelial cells (ECs), we compare ECs derived from bone and skeletal muscle of the same OA patients. OA bone ECs show a pro-inflammatory signature and higher angiogenic sprouting as compared to muscle ECs, in control conditions and stimulated with TNFα. Furthermore, growth of muscle but not bone ECs decreases with increasing patient age and systemic inflammation. Overall, our data demonstrate that inflammatory conditions in OA patients differently affect bone and muscle ECs, suggesting that inflammatory processes increase angiogenesis in subchondral bone while associated systemic low-grade inflammation impairs angiogenesis in muscle, possibly highlighting a vascular trigger linking OA and sarcopenia.

[1] Regenerative Medicine Technologies Lab, Laboratories for Translational Research, Ente Ospedaliero Cantonale (EOC), 6500 Bellinzona, Switzerland. [2] Servizio di Ortopedia e Traumatologia, Ente Ospedaliero Cantonale, Lugano, Switzerland. [3] Euler Institute, Faculty of Biomedical Sciences, Università della Svizzera Italiana, Lugano, Switzerland. [4] Lab of Cancer Genomics, Fondazione "Edo ed Elvo Tempia", Biella, Italy. [5] Laboratory of Biological Structures Mechanics-Chemistry, Material and Chemical Engineering Department "Giulio Natta", Politecnico di Milano, Milan, Italy. [6] NOMIS Center for Immunobiology and Microbial Pathogenesis, Salk Institute, San Diego, CA, USA. [7] IRCCS Ospedale Galeazzi – Sant'Ambrogio, Cell and Tissue Engineering Laboratory, Milan, Italy. [8] Laboratory of Cancer Metastasis Therapeutics, Istituto di Ricerche Farmacologiche Mario Negri-IRCCS, Milano, Italy. ✉email: chiara.arrigoni@eoc.ch

The endothelium, the inner layer of the vascular system, has been increasingly recognized as a heterogeneous tissue, as suggested several years ago from the existence of anatomical differences in endothelia of different organs[1]. Further evidences of endothelial heterogeneity emerged from extensive genomic characterization of organotypic endothelial cells (ECs)[2–4], although originating from mice or fetal human organs. ECs have been indicated as master regulators of tissue homeostasis and regeneration[5] assuming organotypic molecular profiles and functions during development[2,4]. Furthermore, endothelia in different tissues differently respond to external stimuli such as inflammation, whereby the expression of adhesion molecules and leukocyte trafficking in response to inflammatory stimulation is different between different tissues[6]. However, several questions still remain unanswered, such as the inflammatory response of organotypic ECs in pathological conditions or the modifications of EC behavior during aging processes[7,8].

In recent years, the concept of "inflamm-aging", an age-related, low grade systemic inflammation, has been hypothesized as a possible trigger of typical age-related pathologies, including osteoarthritis (OA)[9]. Although OA is considered a joint disease, it has been suggested that low-grade, chronic inflammatory processes occurring locally in OA joints reflect into systemic effects[10]. The presence of an associated systemic inflammation is shown by the slight increase in C-reactive protein (CRP) in OA patients[11]. The rise of inflammatory signaling has been hypothesized to induce vascular endothelial dysfunction and to play a role in the decrease of muscle mass and functioning[12], or even in cerebrovascular dysfunction[10] present in some OA patients, but the underlying mechanisms are not clear yet.

During OA progression, all the joint tissues are subjected to pathological alterations, including the subchondral bone and the synovium, whereby angiogenesis increases. The growth of blood vessels into the osteochondral interface ultimately leads to vascularization of deep cartilage layers and is associated to nerve growth causing pain and joint disfunction[13]. Synovitis (i.e. synovial inflammation) is a well-recognized factor inducing angiogenesis in the synovial membrane, but the role of inflammation in the growth of blood vessels in the subchondral bone is less acknowledged[14]. Conversely, it is known that inflammation damages skeletal muscle microcirculation rather than promoting vessel ingrowth[15] and vascular dysfunction has been recently reported in patients with sarcopenia[16]. Hence, although pathological inflammatory conditions influence blood vessels both in joints and in muscle tissues, the effects of inflammatory mediators on these two organ-specific vascular systems are very different.

In this article we aim at investigating if constitutive differences in organotypic endothelia can cause different functional endothelial responses to inflammatory conditions, which could be at the basis of the alterations in muscle and subchondral bone of OA patients. To this end, we exploited whole-genome expression analysis and microfluidics to evidence genomic and functional differences between patient-matched human ECs, isolated from the subchondral bone and from the skeletal muscle. Differences in growth and response to inflammation were then correlated with OA patient characteristics such as systemic inflammation and age.

## Results

### Systemic inflammation differently affects proliferation of bECs and mECs isolated from OA patients.
We enrolled 31 osteoarthritic patients, and successful isolation of both bone and muscle ECs was achieved for 17 patients, 7 males and 10 females, 5 with grade 2 OA and 12 with grade 3 OA (Table 1). Median age of the patients was 60 years (range 23-76) and median BMI was 24.1,

with values >30 in 3 patients. Median values of C-reactive protein (CRP) and of Erythrocyte sedimentation rate (ESR), indicators of systemic inflammation, were in the ranges considered physiological, with values higher for some obese patients, although CRP and BMI values were not correlated ($r^2 = 0.19$). Following enzymatic isolation and immunoselection, we obtained pure endothelial populations from both muscle and bone tissues. Cells proliferated in culture forming clusters with the typical cobblestone morphology (Figs. 1e and 1i) and expressed endothelial markers such as Vascular Endothelial-cadherin (VE-cadherin), CD31 and Ulex Europaeus Agglutinin -1 (UEA-1) (Fig. 1f–h and 1j–l). We quantified cell proliferation over three passages, which increased exponentially from passage 1 to passage 3 without differences between bECs and mECs (Fig. 1m). Qualitatively, bECs grew sparser than mECs, with less continuous junctions demonstrated by more irregular staining for VE-cadherin, which is internalized also in the cytoplasm (with arrowhead, Fig. 1f). Furthermore, CD31 staining showed evident gaps between adjacent bECs (white arrowhead, Fig. 1g), absent in mEC cultures (Fig. 1j, k). To detect possible influences of patient characteristics on ECs proliferation, we correlated average growth rate values with age and CRP levels (Fig. 1n), showing that the growth rate of both mECs and bECs was negatively correlated with CRP levels in blood ($r^2 = 0.98$ and $r^2 = 0.88$, respectively). The negative correlation in mEC was significantly stronger than in bECs ($p < 0.001$), suggesting that mECs growth was more affected by inflammation levels. Pearson coefficients calculated for the correlation between growth rates and ESR values were similar to those for CRP (Supplementary Fig. 1a), further supporting a negative effect of inflammation on ECs growth. Similarly, both mEC and bEC growth rates negatively correlated with patient age (Fig. 1o), although not significantly ($r^2 = 0.33$ and $r^2 = 0.13$, respectively), with a higher value for mEC as compared to bEC. On the contrary, we found no correlation between mEC and bEC growth with BMI values (Supplementary Fig. 1b) or differences in growth between different OA scores (Supplementary Fig. 1c).

### Differential gene expression reveals a higher expression of inflammatory response genes in bECs as compared to mECs.
We analyzed the transcriptomic profile from 11 patients, comparing patient-matched RNA from bECs and mECs through whole-genome microarrays. We firstly compared gene expression of putative endothelial cells with that of negative cells left after immunoselection, demonstrating a good separation between the populations and thus a high purity of isolated ECs (Fig. 2a). Differential expression analysis (Fig. 2b) revealed that the majority of genes were similarly regulated in bECs and mECs, as reported for other organotypic ECs[3], whilst expression was different for a small subset of genes. In particular, the heatmap in Fig. 2c shows that 38 genes over 56k were significantly upregulated in bECs (including chemokines and apoptosis-related genes), whilst 31 were significantly upregulated in mECs (including genes encoding for ECM proteins, growth factor receptors but also senescence-associated genes). GO-process enrichment analysis highlighted a group of differentially regulated biological processes (Fig. 2d), such as upregulation in bECs of inflammation-related pathways, including response to TNFα (with upregulation of *TNFRSF6B, ICAM1, TNFRSF11B, BIRC3, SELE, PSMB9*) and response to IFNγ (with upregulation of *IFITM1, XAF1, HLA-F, IFI30*). Notably, TNFα is one of the major cytokines involved in OA disease[17], and also IFNγ has been measured in patients with knee OA, weakly correlated with OA grade[18]. Biological processes highly expressed in mECs included ECM organization (upregulation of *LAMA2, FMOD, FBLN5, SULF1, POSTN, MFAP5, LOXL1*) and muscle and nerve cell

**Table 1 patient demographics and clinical parameters.**

| OA Patients | | TOT | Transcriptomic analysis | Angiogenesis assay | Physiological range |
|---|---|---|---|---|---|
| | N | 17 | 11 | 6 | |
| Sex | M | 7 | 5 | 2 | |
| | F | 10 | 6 | 4 | |
| Age [yrs] | Median | 52 | 51 | 64.5 | |
| | Range | 43–76 | 43–76 | 48–72 | |
| BMI [kg/m$^2$] | Median | 24.6 | 25.6 | 24.4 | 18.5–24.9 |
| | Range | 22.1–36.7 | 22.1–36.7 | 22.5–34 | |
| OA grade [Tonnis] | 2 | 5 | 3 | 2 | |
| | 3 | 12 | 8 | 4 | |
| CRP [mg/dl] | Median | 0.15 | 0.19 | 0.105 | 0.08–0.3 |
| | Range | 0.02–0.45 | 0.02–0.45 | 0.06–0.35 | |
| ESR [mm/h] | Median | 8 | 9 | 6.5 | <20(men) |
| | Range | 2–29 | 2–29 | 3–16 | <30 (women) |

Data of OA patients with successful isolation of muscle and bone endothelial cells.

**Table 2 Correlation between GO processes and patient characteristics.**

| Process | NOM $p$-val | | |
|---|---|---|---|
| | age | CRP | BMI |
| Positively correlated | mEC | | |
| Interleukin-12 production | 0.0088 | | |
| NLRP3 Inflammasome complex assembly | 0.0197 | | |
| Reactive oxygen species metabolic process | 0.0320 | | |
| Regulation of cellular extravasation | 0.0440 | | |
| Capillary malformation | | 0.0189 | |
| Negative regulation of myoblast differentiation | | 0.0415 | |
| Negatively correlated | mEC | | |
| Regulation of cell growth involved in cardiac muscle cell development | 0.0138 | | |
| Regulation of response to interferon gamma | | 0.0207 | |
| Positively correlated | bEC | | |
| Cellular response to vascular endothelial growth factor stimulus | | | 0.0018 |
| Positive regulation of sprouting angiogenesis | | | 0.0198 |
| Negatively correlated | bEC | | |
| Negative regulation of tumor necrosis factor superfamily cytokine production | | | 0.015 |
| Positive regulation of interleukin 10 production | | | 0.029 |

GO Processes correlated with patient age, CRP and BMI in mECs and bECs.

between patient age, CRP and BMI and the upregulation of biological processes in both ECs. We found that in mECs some processes, including regulation of inflammatory cytokine production, were positively correlated with age and CRP, whilst processes associated to angiogenesis were negatively correlated. No correlation between angiogenic or inflammatory processes and CRP or age values has been found in bEC, which instead showed a positive correlation of angiogenic processes and a negative correlation of anti-inflammatory processes with BMI (Table 2). To validate the different gene regulation at a protein level, we performed immunofluorescence analyses, finding a higher expression of proteins encoded by bone upregulated genes in bECs (*NOSTRIN* and *SELE*-E, Fig. 2e) and a higher expression for muscle upregulated ones in mECs (*CNGL-1* and *SULF1*, Fig. 2e). Looking at the expression of inflammation-related proteins, we found that E selectin was more expressed in bECs than in mECs, irrespective from patient age and CRP value. On the other hand, TNF receptors such as *TNFSR6b* were more expressed in bECs, although their expression also increased in mECs with increasing patient age (Supplementary Fig. 2). Furthermore, to verify if differential protein expression was present also in the native pathological tissues, we performed immunohistochemical staining in a different set of patient-matched bone and muscle samples, confirming the differential expression of inflammation-related proteins and differentiation factors by endothelial cells also within the tissue (*ICAM-1* and *IGFBP3* Fig. 2F, *TNFSR6b*, *SULF-1* and *HHIP* Supplementary Fig. 3).

**bECs show higher angiogenic sprouting as compared to mECs in a 3D microfluidic sprouting assay.** To verify if differences in gene and protein expression reflected also in functional diversities between bone and muscle ECs, we investigated the angiogenic potential of bECs and mECs ($n = 5$ patients), by monitoring angiogenic sprouting in a microfluidic device (Fig. 3a). Compared to standard sprouting assays, our device allowed to better mimic angiogenic sprouting from existing vessels, being more representative of in vivo conditions. bECs had significantly higher branch number ($p = 0.0038$), area sprout ($p = 0.0012$) and total sprout length ($p = 0.019$) than mECs (Fig. 3b, d, e). To investigate if the OA grade influenced the angiogenic potential, particularly in bEC, we compared the same parameters between grade 2 and grade 3 OA derived bEC and mEC. We found that bEC derived from grade 3 OA patients had a higher branch number and area sprout as compared to bEC derived from grade OA 2 patients, although the differences did not reach statistical significance (Supplementary Fig. 4). On the other hand, bEC branch number was independent from patient CRP values ($r^2 = 0.25$, $p = 0.39$ Fig. 3f), as well as area

differentiation (upregulation of *MYLK2, COL25A1, EPB41L3, ADRB1, RELN, HHIP, IGFBP3*), in agreement with the hypothesis that ECs secrete factors contributing to muscle tissue homeostasis and development[5]. To highlight a possible effect of the pathology on the observed differential gene expression of bECs vs mEC, we compared ECs deriving from patients with different OA grades. We found that there were around 60 genes significantly up or downregulated comparing ECs from patients with OA grade 2 vs. grade 3. Among the genes upregulated in grade 3 ECs as compared to grade 2 ECs, we found cytokines like CCL2 and matrix proteins such as LAMC-2, whilst other genes such as CADH13 were downregulated. Interestingly, only 3 genes, related to IFNγ response, were also differentially expressed in mECs vs bECs, suggesting distinct effects of the tissue of origin and of pathological conditions. We then performed a GSEA (Gene Set Enrichment Analysis) on bECs and mECs to assess a possible correlation

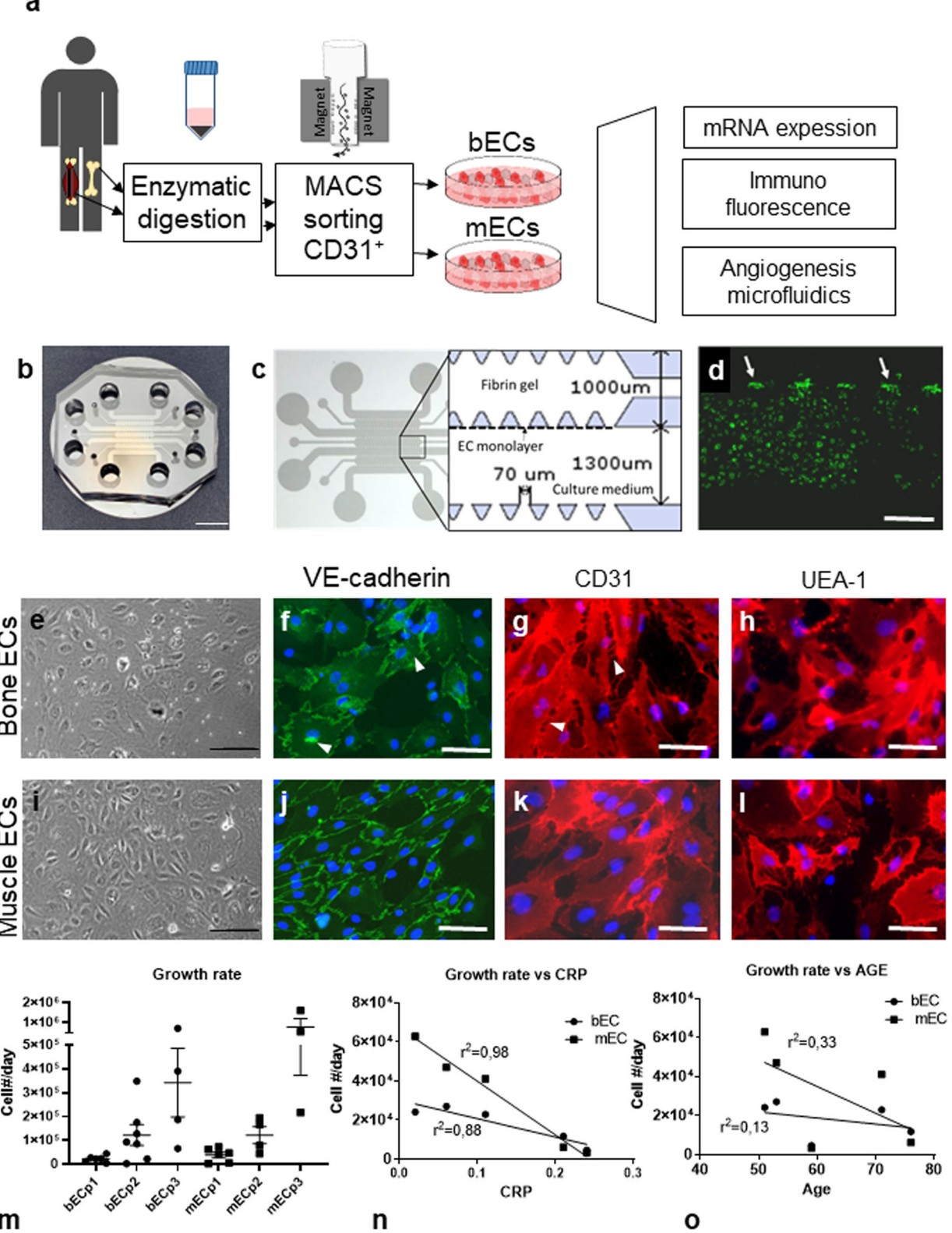

**Fig. 1 Study schematics, microfluidic device images and cell characterization. a** Schematics of experimental procedures. **b** Image of the fabricated microfluidic chip. Scale bar 500 µm. **c** Schematics of the chip channels. **d** Fluorescence image of a monolayer of endothelial cells on the chip channel (green). White arrows indicate the areas in which the monolayer contacts the fibrin gel. Scale bar: 250 µm. **e–l** Characterization of cultured bECs and mECs: **e–i** Phase contrast images. Scale bars 100 µm. **f–j** VE-cadherin staining (green, blue: DAPI). **g–k** CD31 staining (red, blue: DAPI). **h–l** UEA-1 staining (red, blue: DAPI). Scale bars 50 µm. **m** Growth rate of bECs and mECs over the first three passages ($n = 3$–6). Data are represented as mean and SEM. **n** Correlation between growth rate at the first passage and CRP values of the patients ($n = 5$, $p = 0.0187$ and $p = 0.001$ for bECs and mECs respectively). **o** Correlation between growth rate at the first passage and patient age ($n = 5$, $p = 0.55$ and $p = 0.30$ for bECs and mECs respectively).

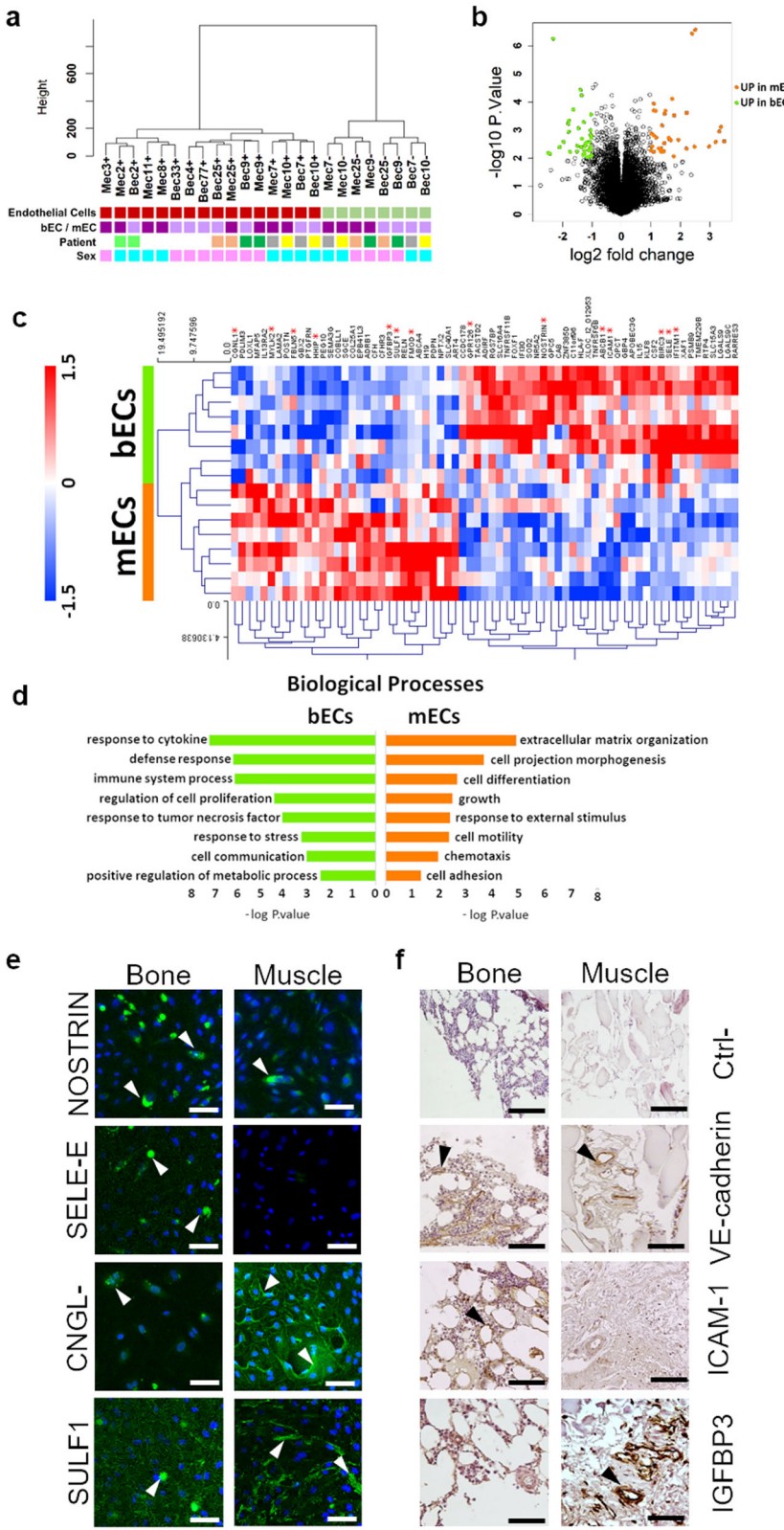

sprout, total sprout length and average branch length, altogether suggesting absence of correlation between angiogenic sprouting and systemic inflammation for bone derived ECs. On the contrary, an almost significant negative correlation was observed for the sprouting of mECs, (as shown in Fig. 3C: $r^2 = 0.67$, $p = 0.08$), suggesting a lower angiogenic potential in mECs derived from patients with higher systemic inflammation (Fig. 3f).

**Inflammatory stimulation exerts different effects on bECs and mECs**. To further investigate if different angiogenesis between mEC and bEC was related to a different response to inflammatory conditions, we stimulated bECs and mECs with TNFα or IFNγ in the microfluidic chip, quantifying the resulting sprouting angiogenesis. We firstly verified through computational simulations that we were able to generate a stable gradient of cytokines in the

**Fig. 2 transcriptomic analysis, immunofluorescence and immunohistochemistry showing a differential gene expression between bECs and mECs.**
Results of microarray transcriptomic analysis. **a** Clustering analysis on all isolated cells, i.e. putative endothelial cells from bone (bECs + ) and muscle (mECs + ) and co-isolated non endothelial cells from the same tissues (bEC- and mEC-). Data were grouped following the cell type (endothelial cells: red squares, non-endothelial cells: green squares), the tissue of origin (bone: bEC, light purple squares, muscle: mEC, dark purple squares), the patient of origin and the sex of the patient (males: light blue squares, females: pink squares). **b** Volcano plot showing differences in the expression of genes between bECs and mECs. Black dots represent genes not significantly different, green dots represent gene significantly upregulated in bECs, and orange dots represent genes more expressed in mECs. **c** Heatmap showing the differentially expressed genes ($p < 0.01$, log2FC >1) between bECs and mECs. Asterisks represent genes analyzed also at a protein level. **d** GO-process analysis showing the main biological processes significantly upregulated in bECs (green) or in mECs (orange). **e** Immunofluorescence staining of cultured bECs and mECs. Green staining represents the protein (NOSTRIN, CNGL-1, SELE-E, SULF-1), nuclei in blue (DAPI). White arrowheads point at the positive signal. Scale bars 50 μm. **f** Immunohistochemical staining of bone and muscle tissues. Brown staining represents the protein (VE-cadherin, to identify vessels position, ICAM-1 and IGFBP-3), hematoxylin counterstaining in violet. Black arrowheads indicate positive areas. Scale bars 100 μm.

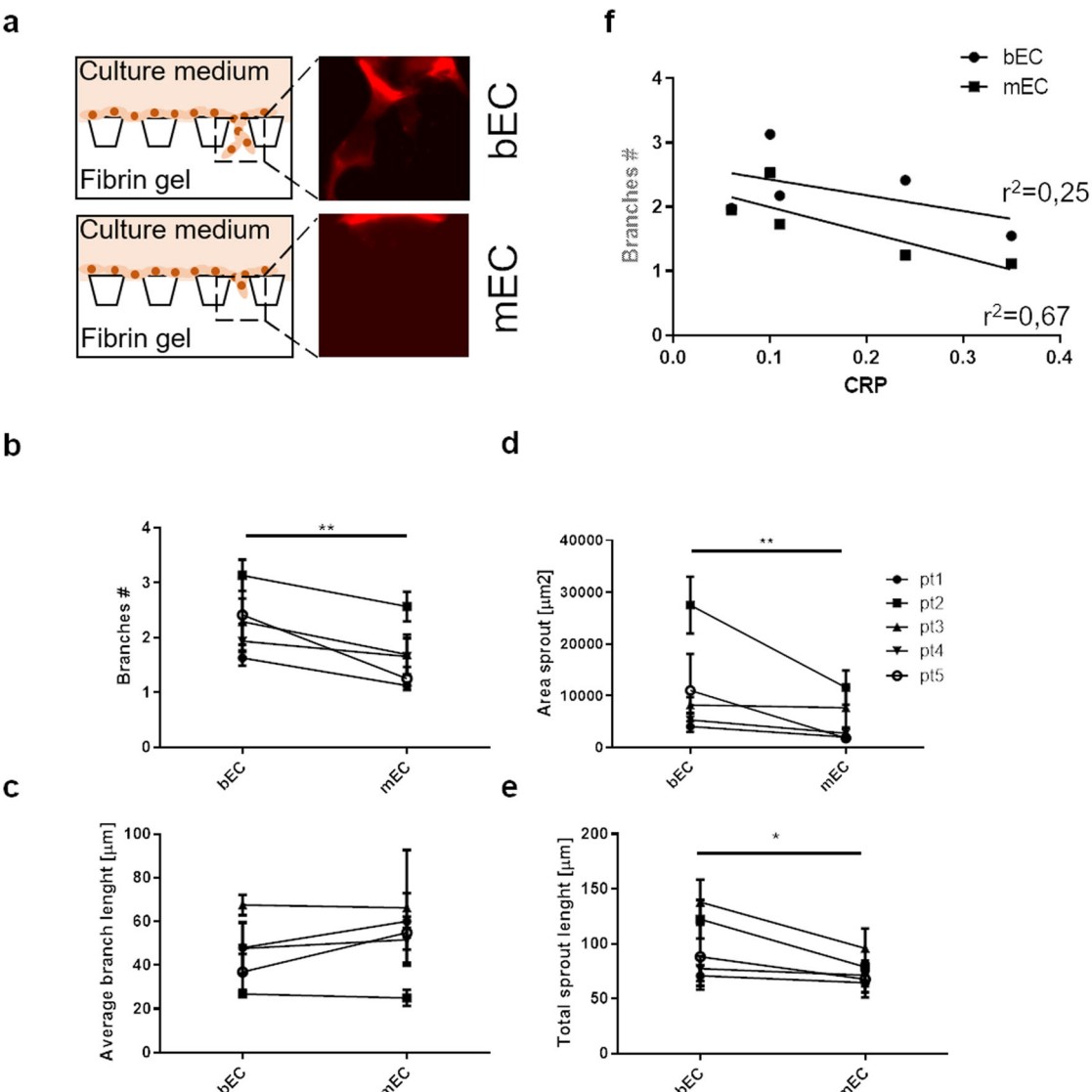

**Fig. 3 Microfluidic sprouting assay to quantify angiogenic behavior of bECs and mECs in control conditions. a** Schematics of the angiogenesis assay, showing the channel in which the ECs were seeded (upper part), separated by a row of posts from the lower channel containing fibrin. Images on the right show representative immunofluorescence images of bECs and mECs invading the fibrin channel. Paired comparison of **b** branch number, **c** average branch length, **d** area sprouting and **e** total sprout length between bECs and mECs of single patients ($n = 5$, *$p < 0.05$; **$p < 0.01$). Average and SEM of at least 5 replicates per patient are shown. **f** Interpolation of the branch number plotted against CRP values ($n = 5$), showing a stronger negative linear correlation in mECs ($r^2 = 0.67$, $p = 0.08$) as compared to bECs ($r^2 = 0.25$, $p = 0.39$).

chip during the experimental time (Fig. 4a). Graphs in Fig. 4b–e report the values for angiogenic parameters normalized to the value of bECs in control conditions. TNFα addition increased branch number (Fig. 4b) and area sprout (Fig. 4c) in bECs but not

in mECs, whose values remained significantly lower than bECs. Total sprout length (Fig. 4e), increased both in bECs and in mECs with TNFα addition. This higher length of the total sprouting was due to an increased value of the average length of each angiogenic

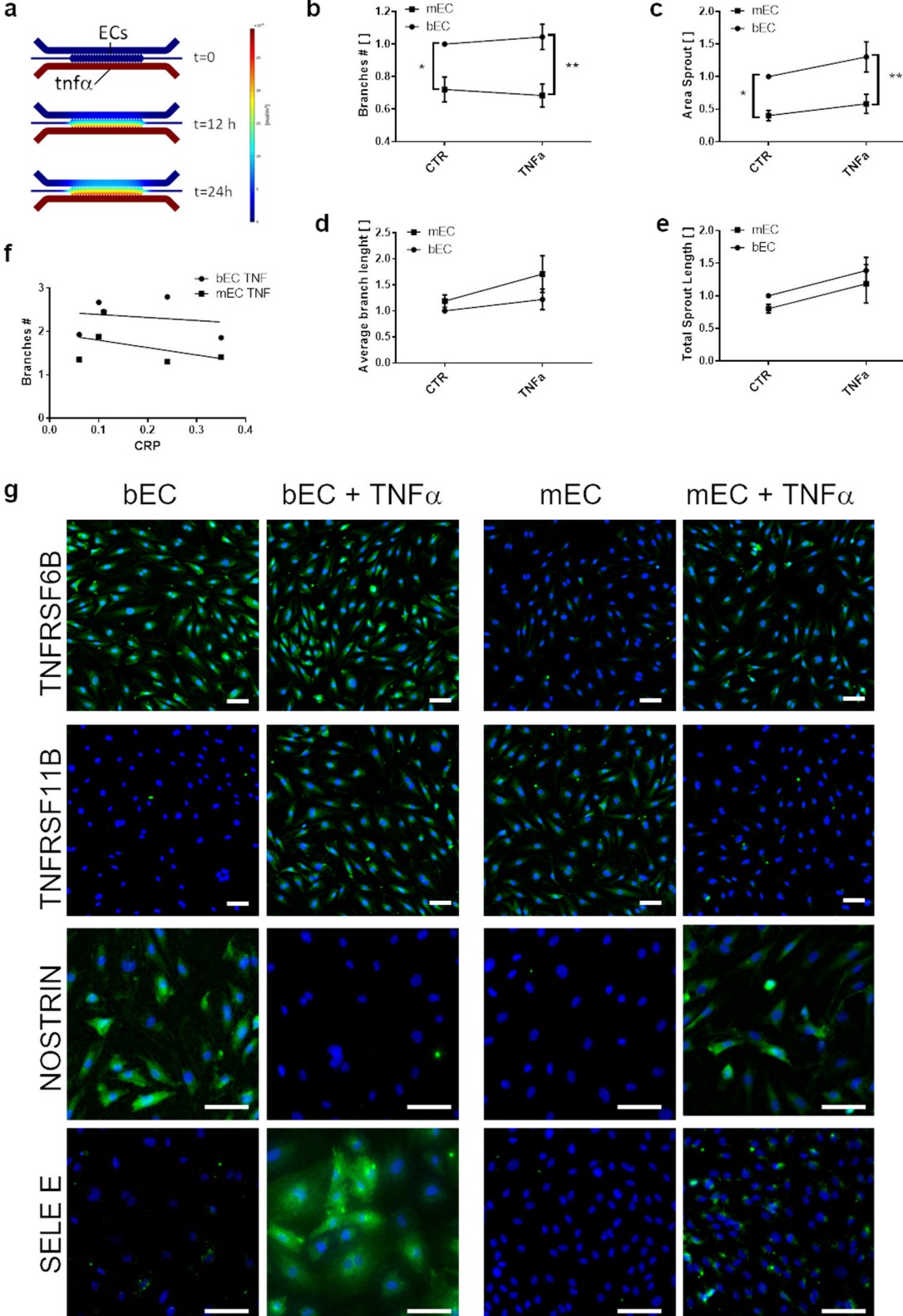

**Fig. 4 angiogenic behavior and marker expression of bECs and mECs under TNFα stimulation. a** Results of the computational simulation showing the establishment of a stable gradient of TNFα after 24 h of culture in the chip. Graphs comparing values of **b** branches number, **c** area sprouting, **d** average branch length and **e** total sprout length for bECs and mECs in control conditions and under stimulation with TNFα. Data were normalized on the value of bECs in control conditions, which was set at 1 ($n = 5$, *$p < 0.05$; **$p < 0.01$). Average and SEM of at least 5 replicates per patient are shown. **f** Interpolation of the branch number plotted against CRP values, for mECs and bECs under TNFα stimulation, showing no correlation. **g** Immunofluorescence staining of bECs and mECs in control conditions or stimulated with TNFα. Green staining represents the protein (TNFRSF6b, TNFRSF11, NOSTRIN, SELE-E), nuclei in blue (DAPI). Scale bars 50 µm.

branch in mEC after TNFα addition (Fig. 4d), and not to the increase in branch number as observed in bECs (Fig. 4b). We found no correlation between branch number of TNFα–stimulated bECs and mECs with CRP levels (Fig. 4f). Furthermore, inflammatory conditions differently modulated the expression of TNF receptors and adhesion molecules in mECs and bECs. SELE-E, one of the major drivers of leukocyte adhesion and extravasation during inflammation[19], was weakly expressed in control conditions in bECs but not in mECs, and was evidently upregulated after TNFα addition only in bECs. Similarly, TNFα addition in bECs, but not in mEC, upregulated the expression of TNFRSF11B, whose increased serum levels have been found in patients with inflammatory pathologies such as diabetes or atherosclerosis[20]. TNFRSF6b maintained a stable expression in bECs and was slightly induced in mECs after TNFα addition. Interestingly, NOSTRIN, a negative regulator of nitric oxide production with antiangiogenic activity[21], decreased in bECs and increased in mECs after TNFα stimulation, in agreement with angiogenesis assay results (Fig. 4g). The pro-angiogenic effect of inflammatory cytokines in bECs was specific for TNFα, since addition of IFNγ gave opposite results, decreasing bECs and increasing mECs sprouting (see Supplementary Fig. 5b). A negative correlation trend, although not significant, was observed between angiogenic sprouting induced by IFNγ in mEC and CRP levels (Supplementary Fig. 5c), possibly related to the observed decreased response to IFNγ with increasing CRP values observed in mEC (Table 2).

## Discussion

In this work we combined microarray-based transcriptomics and microfluidics to compare ECs derived from bone and skeletal muscle of the same OA patient, showing differential regulation of genes and biological processes, which reflected into different EC behaviors. Although single cell sequencing would have provided a more comprehensive data set, we opted for microarray analyses since low transcripts genes may have gone undetected using the former technology. Our results show that in OA patients bone ECs are primed in a chronically inflamed state, as demonstrated by the upregulation of processes related to inflammation and response to cytokines. This transcriptional signature is maintained after in vitro culture, contrarily to previously shown data on tonsil-derived ECs[22], resulting in a different phenotype as compared to skeletal muscle-derived ECs. One limitation of our study is related to the unavailability of healthy subject-matched bone and muscle endothelia as control conditions, thus, to highlight possible effects due to the pathology, we compared ECs from patients with different OA degrees, although being all end-stage disease patients. We found that the genes differentially expressed between mEC and bEC were not shared with those differentially expressed between OA grade 2 and grade 3 ECs. Furthermore, when we compared the transcriptomic profile of OA ECs with that of healthy HUVECs[23], we did not find possible signatures induced by OA in both ECs as compared to a healthy endothelium. Taking these considerations together, we can suppose that the differential expression observed between OA bECs and mECs is mainly attributable to the different origin of endothelial cells, as widely reported in the literature for other healthy tissue types and organisms[2,4]. In particular, the higher expression of adhesion molecules found in OA bECs as compared to OA mECs, is in agreement with the already reported specific over-expression of SELE-E in healthy bone endothelium[24]. Interestingly, among the very few genes which had a differential regulation between OA grade 2 and 3 and that were different also between bECs and mECs, there were genes involved in the response to IFNγ. In particular, OA grade 3 bECs showed

significantly reduced IFITM-1 expression as compared to grade 2 bECs, suggesting that the response to IFNγ could be also influenced by pathological conditions beside tissue-specificity, according to literature results showing that OA reduces the expression of IFNγ receptors in articular cells[25].

After analyzing gene expression, we also compared the angiogenic capability of the two different ECs, showing that bECs originated more branched and extended sprouting as compared to mECs. Furthermore, angiogenic sprouting of bECs slightly increased with OA severity, accordingly with the phases of disease progression, which involves increasing blood vessel growth in the subchondral bone at final OA stages[26]. From the results of our study, increased vessel formation seems more attributable to increased angiogenic sprouting than to increased cell proliferation, since no difference in bEC growth between grade 2 and 3 has been evidenced. To further investigate possible factors associated to the increased bEC angiogenesis, we analyzed their behavior respective to inflammation. We found that bECs were quite insensitive to CRP values, both in terms of proliferation and angiogenic potential, supporting the absence of correlation between the expression of angiogenic processes in bECs and CRP. However, when directly stimulated with TNFα, one of the major pro-inflammatory cytokines present in OA joints[27], bEC showed an increased angiogenic potential, suggesting that local but not systemic inflammation could indeed play a role in subchondral bone angiogenesis during OA. The pro-angiogenic role of TNFα on the subchondral bone blood vessels has been reported also in a mouse model of OA, and was mediated by the production of LRG1, a glycoprotein modulating pathological angiogenesis[26]. We also found a positive correlation of angiogenic processes in bEC with patient BMI. As reported in the literature, BMI is a risk factor for OA[28] and indeed patients with more severe OA had higher BMI values (22.8 ± 0.9 vs 28.5 ± 5.5 respectively, for grade 2 and 3 OA). However, in our patient subset, BMI was not correlated to CRP values, thus making implausible that the pro-angiogenic effects of increased BMI are also associated to higher inflammation, whilst suggesting a possible further relation with the altered mechanical loading[29].

Regarding muscle endothelium, mEC showed a decrease in growth and angiogenic sprouting with increasing CRP values, further supported by the non-increased angiogenic potential of mECs directly stimulated with TNFα. Moreover, expression of IGFBP3, marker of senescent ECs[30], was higher in mECs deriving from patients with higher CRP values and further increased with TNFα stimulation (Supplementary Fig. 6), overall indicating that inflammation can damage muscle endothelium. This detrimental effect of inflammation on mECs can lead to vascular bed damage, possibly contributing to skeletal muscle loss observed in OA patients[12]. Interestingly, sarcopenia is observed mainly in OA patients presenting high CRP values[31] and has not been related to the decrease of other factors critical for muscle homeostasis such as satellite cell count[32].

The effects of TNFα and IFNγ on endothelial cells angiogenesis have been widely studied in the literature, with the majority of studies describing a pro-angiogenic role of TNFα[33,34], whilst less consensus exists on the effects of IFNγ, which have been described as either pro-angiogenic[35] or anti-angiogenic[36], with a detrimental effect on bone endothelium[37]. In our study, IFNγ effects were opposite to those of TNFα, increasing angiogenesis in mECs while decreasing it in bECs, further evidencing how angiogenic response to cytokine stimulation can be differentially modulated in different endothelia. These results extend those of previous studies on the response of different ECs (HUVECs and HAECs) to cytokines[38] and well fit with the hypothesis of the existence of an organ-specific endothelial inflammation signature, recently shown in the mouse[39].

Finally, the effect of age was almost negligible for bEC, whilst it was negatively correlated with growth and positively correlated with inflammatory processes in mECs. Since age was not correlated with CRP levels, it can be indicated as another independent detrimental factor for muscle endothelium, according to other reports in the literature[40], although more data are needed to support its contribution on skeletal muscle vascular damage.

In conclusion, our results show that ECs deriving from bone and skeletal muscle of the same human subject present genomic and phenotypic differences, as previously demonstrated for ECs originating from different mouse organs. Furthermore, bone ECs were more angiogenic than muscle ECs, in a higher extent for more severe OA, prompting that local inflammatory processes in OA joints progressively lead to increased angiogenesis in subchondral bone. On the other hand, lower angiogenesis in muscle endothelial cells suggests that OA-associated systemic inflammation can damage muscle vascular bed, possibly being at the basis of sarcopenia observed in OA patients with high inflammation levels.

## Methods

**Cell isolation and culture**. Primary Ecs were isolated from leftover material (bone and muscle tissues) deriving from patients subjected to hip replacement surgery ($n = 31$), who gave informed consent. The study has been approved from Hospital san Raffaele Ethical Committee (EC opinion register number: 50/INT/2015, date 01/04/2015, registration number from ClinicalTrials.gov: NCT04047459). After surgical aseptic removal, bone tissue has been fragmented in small pieces (around 2 mm) with tweezers and a scalpel, before being digested in a solution of collagenase type I (Worthington, 2 mg/ml) in DMEM, for 1 h at 37 °C in an orbital shaker. Muscle tissue was fragmented in small pieces (around 1 mm) with tweezers and scissors, then it was digested with an enzymatic mix solution (Skeletal muscle dissociation kit, Miltenyi Biotech), for 1 h at 37 °C in an orbital shaker. After digestion, enzymes were inactivated by adding an equal volume of DMEM with 10% FBS, bone and muscle lysates were filtered through a 100 μm cell strainer and resuspended in a solution for red blood cells lysis (Red Blood Cell Lysis Solution, Miltenyi Biotech) for 2 min. Cell suspensions were centrifuged at 1200 rpm for 3 min, resuspended in complete endothelial growth medium (EGM-2, Lonza), plated on T25 flasks coated with 10 μg/ml fibronectin (Merck Millipore) and left in culture until clones of endothelial morphology started to appear. Cells were then harvested and putative Ecs were identified through immunomagnetic selection for CD31+ cells (MACS CD31 kit, Miltenyi Biotech)[41]. At least two rounds of immunomagnetic selection have been carried out to avoid the presence of cells of non-endothelial phenotype. After a passage in culture with EGM-2 medium (Lonza), muscle (mECs) and bone (bECs) Ecs were partly seeded for immunofluorescence on 96 well microplates (Ibidi), partly frozen for RNA extraction and a last aliquot was conserved for subsequent experiments, as reported in Fig. 1a.

**Gene expression analysis**. RNA was extracted following standard procedures (PureLink RNA Mini Kit, Ambion). Briefly, cells were lysed in lysis buffer, loaded into spin cartridge columns; obtained RNA has been digested with Dnase (PureLink, Qiagen) and washed from excess ethanol. RNA quality and quantity were assessed using Agilent 2100 bioanalyzer (Agilent Technologies, Santa Clara, CA) and NanoDrop ND-1000 Spectro-photometer (Thermo Fisher Scientific, Waltham, MA), respectively. Gene expression profiling was carried out using the one-color labeling method: labeling, hybridization, slide washing and scanning were performed following the manufacturers protocols (Agilent Technologies). Briefly, mRNA from 100 ng of totRNA was amplified, labeled with Cy3 and purified with columns. 600 ng of labeled specimens were hybridized on Agilent Human Gene Expression v3 8x60K microarrays. After 17 h, slides were washed and scanned using the Agilent Scanner version C (G2505C, Agilent Technologies). Images were analyzed using the Feature Extraction software v10.7.3.1 (Agilent Technologies). Raw data elaboration was carried out with Bioconductor (www.bioconductor.org)[42], using The LIMMA (Linear Models for Microarray Analysis) R package. Background correction was performed with the *normexp* method with an offset of 50, and *quantile* was used for the between-array normalization. The empirical Bayes method was used to compute moderated t-statistics for two-class paired tests[43].

Transcripts with an absolute log₂ fold change (logFC) in bECs vs mECs >1 and adjusted *p*-value < 0.01 were considered as differentially expressed.

Over-represented biological processes of the Gene Ontology (GO) were investigated with the functional annotation tool available within DAVID Knowledgebase v2022q3 (https://david.ncifcrf.gov/). Gene Set Enrichment Analysis (GSEA) was used to evaluate significant enrichment in predefined sets of genes (https://www.gsea-msigdb.org/gsea/).

**Immunohistochemistry and immunofluorescence analyses**. To validate the differential expression at the protein level, we performed immunohistochemical analyses on patient-matched muscle and bone tissue samples and immunofluorescence analyses on cultured bECs and mECs, following standard immunohistochemistry and immunofluorescence protocols. The following antibodies were used for both analyses: human VE-cadherin (mouse monoclonal anti CD144, Thermofisher, 1:50), human ICAM-1 (mouse monoclonal anti-rat ICAM-1, Thermofisher, 1:200), human IGFBP-3 (rabbit polyclonal anti human IGFBP3, Thermofisher, 1:50), human CD31 (recombinant anti CD31 (biotin), AbCam, 1:200), UEA-1 (Biotinylated Ulex Europaeus Agglutinin I, Vector Labs, 1:200 for IF), NOSTRIN (rabbit polyclonal anti-NOSTRIN Thermofisher 1:500), CNGL-1 (rabbit polyclonal anti-human CNGL-1, 1:1000), E-selectin (mouse monoclonal anti CD62E, Thermofisher, 1:500), SULF-1 (rabbit polyclonal anti-human SULF-1, Thermofisher, 1:500), TNFRSF6B (rabbit polyclonal anti human DcR3, Thermofisher, 1:100), TNFRSF11B (mouse monoclonal antiOPG, Thermofisher, 1:200).

Bone and muscle tissue fragments were fixed for 24 or for 5 h, respectively, in 4% neutral-buffered formalin, washed in tap water and demi-water and then processed for histology. Before processing, bone fragments were decalcified in Morse's solution from 5 to 7 days depending on tissue dimensions. Samples were dehydrated through a graded series of ethanol and embedded in paraffin. 5-μm-thick sections were cut and mounted on Thermo Scientific™ SuperFrost Plus™ glass slides. Slides were deparaffinized, rehydrated and exposed to citrate antigen retrieval. Slides were washed with washing buffer (1% PBS + Tween20 1:2000), washed with hydrogen peroxide to quench endogenous peroxidases, blocked with 5% bovine serum albumin (BSA, Sigma Aldrich) for 30 min, incubated for 1 h with primary antibodies in 5% BSA, washed, incubated with biotinylated secondary antibody (anti-(species of primary host), Vector Laboratories, 1:200 in PBS), washed, incubated for 30 min with peroxidase (HRP)-conjugated ABC reagent (Vectastain ABC kit, Vector Laboratories, 1:150), washed and incubated for 8 min with DAB (ImmPACT® DAB Substrate, HRP, Vector Laboratories, 1 drop of DAB Stock Solution for 1 ml of diluent). Finally, slides were washed with tap water, counter stained for 1 min with Hematoxylin, dehydrated and mounted. Histological sections were imaged with a Nikon Eclipse Ti microscope. Cultured cells were fixed with 4% paraformaldehyde, permeabilized with 0.1% Triton X-100 for 10 min and treated with 4% BSA for 1 h. Primary antibodies diluted in 5% BSA were incubated at at 37 °C for 1 h and then washed twice in PBS. Secondary antibodies (goat Alexa Fluor 488 conjugated anti-mouse IgG, Thermofisher1:2000; goat Alexa Fluor 488 conjugated anti-rabbit IgG, Thermofisher1:500; goat Alexa Fluor® 568 conjugated anti-rabbit IgG, Thermofisher, 1:1000) and 300 nM DAPI (4'6-Diamidino-2-Phenylindole) diluted in PBS were added to the samples and incubated at 4 °C overnight. Samples were imaged with an Olympus IX71fluorescence microscope.

**Microfluidic chip fabrication**. The microfluidic chip has been fabricated as previously reported[44] and consists of three hydrogel regions each flanked by two lateral media channels (Figs. 1b and 1c), separated by trapezoidal posts to allow the confinement of the hydrogel matrix. Briefly, a silicon wafer with the desired geometry was produced through standard soft lithography techniques (FlowJem Inc.), then poly-dimethyl-siloxane (PMDS) was poured on the wafer and cured in an oven at 80 °C for 2 h. Once polymerized, the PDMS was removed from the wafer, reservoirs were created with biopsy punches (2 mm diameter) and the PDMS was attached to a glass coverslip through plasma bonding.

**Computational simulation of chemokine gradient formation**. To quantify TNF-α and IFN-γ diffusion within the hydrogel matrix and the generation of a chemoattractant gradient, two dimensional simulations were performed using the finite element method (FEM) software Comsol Multiphysics. The lateral channel adjacent to the endothelial monolayer was considered to be filled with medium for cell culture, while the opposite channel was assumed to contain the same medium supplemented with 5 ng/ml TNF-α or 50 ng/ml IFN-γ. This lateral channel was considered as chemoattractant source and diffusion was simulated over 24 h (time-dependent solver: generalized-α; time step: 1 h). Zero mass flux was considered at the device boundaries. The diffusion coefficients of TNF-α and IFN-γ were assumed equal to $1.49 \times 10^{-7}$ cm²/s and $1.25 \times 10^{-7}$ cm²/s, respectively[45]. The results of the simulations show that a chemoattractant gradient for both TNF-α and IFN-γ was maintained through the extracellular matrix (ECM) channel for at least 24 h. These simulated results were then used to setup our experimental protocol.

**Angiogenesis assay on chip**. Fibrinogen at 5 mg/ml in EGM-2 medium was mixed with thrombin at 4 U/ml in the same medium and injected in two non-adjacent gel channels. The gel was left to polymerize at 37 °C for 20 min. After polymerization, a suspension of endothelial cells at 3 M cells/ml has been injected in the two external medium channels and the microfluidic chips were left at 37 °C to allow endothelial cell adhesion to fibrin for 30 min. We then added 50 μl of EGM-2 medium and left the chip in culture for 4 days, until an endothelial cell monolayer formed between the chip pillars (Fig. 1D). For experiments about the effects of inflammatory conditions, TNFα and IFNγ have been added to endothelial

cell monolayers in multiwell plates or in the central medium channel of microfluidic chip at a concentration of 5 ng/ml[46] and 50 ng/ml[47], respectively.

**Imaging and angiogenesis quantification**. After 4 days of incubation in culture medium with or without inflammatory cytokines, microfluidic chips (at least 3 chips for each patient) were imaged (at least 2 ROI per chip) under a fluorescence microscope (Olympus IX71). Tiff images were then processed with Image J to quantify angiogenesis parameters. In particular, we firstly binarized the.tiff image applying automated thresholding, then we applied Image J plugins skeletonize and analyze skeleton, to calculate all the morphological parameters of the vascular sprouting in the image (number of branches, length of each branch and total area of the sprouting). Total sprouting length and average branch length were calculated as the sum and the average of all the lengths of the single branches, respectively. For angiogenesis in inflamed conditions, quantified values of angiogenic sprouting parameters of bECs and mECs in presence of the two cytokines were normalized to control conditions values.

**Statistics and reproducibility**. Data were analyzed using GraphPad Prism 8 (Graphpad Software Inc.). For correlation analyses, Pearson coefficient $r^2$ and $p$-value have been calculated using the average value of at least 5 replicates for each patient, whereby more than one replicate was available (e.g. for angiogenic parameters). For the correlations between ECs growth rate and CRP, age and BMI 5 patients were included, whilst for correlation between ECs growth rate and ESR, 4 patients were considered. Furthermore, regarding the correlations between angiogenic and patient parameters, we considered 5 patients in control and TNFα stimulated conditions and 4 patients for IFNγ effect. To test differences in growth rate between different ECs or different OA grades, standard two-ways ANOVA test with Tuckey post-hoc test was used. At least 4 samples (corresponding to different patients) for each passage and ECs were considered in the comparison of growth rate between bECs and mECs, whilst 2 and 3 samples were included in the comparison between grade2 and grade3 OA (both for growth rate and for and angiogenic parameters), respectively. The differences in angiogenic parameters between bECs and mECs in control conditions and in inflamed conditions (both TNFα and IFNγ) were tested through a paired two-way ANOVA with Sidak post hoc test. For control and TNFα stimulated conditions five patients were included whilst for IFNγ four patients were considered. Differences have been considered significant with ***$p < 0.001$, **$p < 0.01$, *$p < 0.05$.

**Reporting summary**. Further information on research design is available in the Nature Portfolio Reporting Summary linked to this article.

## Data availability

Raw and processed transcriptomic data were deposited on the GEO Omnibus database (GSE172419). Raw numeric data used for generating the graphs were deposited in OSF (https://osf.io/63nma/?view_only=d9aa5e5499704d1b860c40e79a59f6e2). Other data are available from the corresponding author on reasonable request.

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

## Acknowledgements

This work was supported by the Office of the Assistant Secretary of Defense for Health Affairs through the Breast Cancer Research Program under Award No. W81XWH-15-1-0092. Opinions, interpretations, conclusions and recommendations are those of the authors and are not necessarily endorsed by the Department of Defense.

## Author contributions

Conceptualization, C.A., S.B., C.G., M.R.B and M.M.; transcriptomic analyses P.O., I.G., M.M.-G. and G.C.; cell isolation and culture: C.A., M.G., M.C. and M.V.C.; microfluidic assay: M.C., immunohistochemistry and immunofluorescence: M.V.C; surgical sample collection and scoring: L.Z.; writing—original draft, C.A. and S.B.; writing—review and editing, P.O., M.G., C.G., M.R.B., G.C. and M.M.; funding acquisition, and C.C and M.M.; supervision, M.M.

## Competing interests

The authors declare no competing interests.
