## [Peer Review File · Communications Biology]

Reviewers' comments:

Reviewer #2 (Remarks to the Author):

The paper investigated organ-specific features of endothelial cells (ECs) derived from osteoarthritis (OA) patients and their association with OA and sarcopenia. The authors found that bone-derived ECs (bECs) exhibit pro-inflammatory characteristics compared to muscle-derived ECs (mECs), whereas mECs are more susceptible to systemic inflammation. The authors' findings may lead to a better understanding of the pathogenesis of OA complicated by sarcopenia. However, the following points need to be addressed or substantiated.

1) A major limitation of this study is the lack of healthy control subjects. Therefore, it is unclear whether the described organ-specific differences in ECs are due to OA or to differences in intrinsic characteristics. I understand that it is difficult to obtain samples from healthy humans. However, comparison with a healthy sample is essential to understanding the pathological mechanisms. Therefore, it is strongly recommended to discuss this point in the discussion section at least as a limitation of this study.

2) In Table 1, patient information such as grade of OA and degree of sarcopenia (grip strength and muscle mass) should be provided.

3) In the legend of Figure 1P and Q, the number of patients is described as n=6, but the dots in the graph are n=5. In supplementary figure 1, the dots on the graph is n=4. Please correct them.

4) In Figure 2A, please provide a description of the columns below the clustering diagram, such as "bEC/mEC" or "Patient"

5) In the legend of Figure 2, labels "B" and "C" are interchanged.

6) In Table 2, why are only the GSEA results for mECs shown, even though the GO analysis shows results for upregulation of inflammatory pathways in bECs (Figure 1D)? Please provide convincing explanations.

7) The authors showed that different responses of ECs to TNF α or IFN γ (Figure 4 and supplementary figure 4). Interpretations of the different responses of ECs to inflammatory cytokines should be discussed in the discussion section.

Reviewer #3 (Remarks to the Author):

In the present study, the authors explored if constitutive differences in organotypic endothelia can cause different functional endothelial responses to inflammatory conditions, which could be at the basis of the alterations in muscle and subchondral bone of osteoarthritis patients. They used an approach combining microfluidic angiogenic assay with genome-wide transcriptome profiling to evidence genomic and functional differences between human endothelial cells, isolated from the subchondral bone and from the skeletal muscle of the same patient. Then, they correlated differences in growth and response to inflammation with osteoarthritis patient characteristics such as systemic inflammation parameters, as CRP, and age.

Osteoarthritis induced a pro-inflammatory gene signature in bone derived endothelial cells compared to patient-matched muscle endothelial cells, with an upregulation of processes like cytokines' response and expression of adhesion molecules and TNF- α receptors. Angiogenic sprouting of muscle endothelial cells was lower than that of bone endothelial cells in control conditions and after TNF- α

stimulation. Besides, growth of muscle but not of bone endothelial cells decreased with increasing age and systemic inflammation. The obtained findings indicate that inflammatory conditions in osteoarthritis patients differently affect bone and muscle endothelial cells. Specifically, inflammatory processes possibly increase angiogenesis in subchondral bone and associated systemic low-grade inflammation could impair angiogenesis in muscle. Interestingly, the whole data also provide an explanation for the sarcopenia often observed in osteoarthritis patients with high inflammation levels.

Collectively, the presented data are of good quality and experiments are well designed. The manuscript is quite well written and only needs of few corrections and improvements:

- Page 9: The authors should specify they enrolled osteoarthritis patients both in the text and in the Table.
- Figure 1D should be quoted within the manuscript.
- The use of microarray as strategy to obtain differential gene expression data should be mentioned in the text of both Results and related Figure legend.
- Legend to Figure 2: descriptions of panels B and C are inverted and must be corrected.
- Discussion: The authors could improve this section by examining the limitations of their study; for instance, they used microarrays to analyze the whole gene expression instead of NGS strategies (especially single cell transcriptomics), which could provide further insights on the knowledge of "inflamm-aging" and related pathologies if combined with microfluidics.

Reviewer #4 (Remarks to the Author):

The authors describes the finding of OA patients (17) samples that shows that inflammatory processes increase angiogenesis in subchondral bone while associated systemic low-grade inflammation impairs angiogenesis in muscle. These results are derived mainly from gene expression, immunohistochemistry and angiogenesis assays.

Overall the findings of the study are interesting and the experiments are technically sound. I am not an expert in the research field so won't comment on the impact and significance of the work. I have following technical concerns, which need to address before accepting the manuscript for publication.

Major:

1. Inflammation is directly linked with the BMI and if the 17 samples are analyzed based on group of low, high and very high BMI, It is advisable that gene expression data also re-analyzed in groups of BMI values. Although the sample size is small here, it might indicate the correlation of BMI and inflammation linked markers.

2. Discussion of the manuscript is not written clearly. Authors should put forward the prospective of their study in the field

3. Immunofluorescence data quality lack clarity

1. All Figures: scale bar is not visible

2. S Figure 5: bar is not there and figure legend is not clear, CRP values should be 0.3 not 0,3, similar typo mistake are identified several places.

3. Figure 4B panel is not clear in reading.

Reviewers' comments:

Reviewer #2 (Remarks to the Author):

Comment	Answer
The paper investigated organ-specific features of endothelial cells (ECs) derived from osteoarthritis (OA) patients and their association with OA and sarcopenia. The authors found that bone-derived ECs (bECs) exhibit pro-inflammatory characteristics compared to muscle-derived ECs (mECs), whereas mECs are more susceptible to systemic inflammation. The authors' findings may lead to a better understanding of the pathogenesis of OA complicated by sarcopenia. However, the following points need to be addressed or substantiated.	We thank the reviewer for the kind appreciation of our work and for its precious comments
A major limitation of this study is the lack of healthy control subjects. Therefore, it is unclear whether the described organ-specific differences in ECs are due to OA or to differences in intrinsic characteristics. I understand that it is difficult to obtain samples from healthy humans. However, comparison with a healthy sample is essential to understanding the pathological mechanisms. Therefore, it is strongly recommended to discuss this point in the discussion section at least as a limitation of this study.	We acknowledge that the point raised by the reviewer is at the same time crucial and really difficult to approach rigorously. Indeed, we cannot isolate matched bone and skeletal muscle ECs from healthy subjects for ethical and technical reasons. We thus looked for the presence of transcriptomic data in the literature comparing bone and muscle human endothelia. Unfortunately, there are no publicly available gene expression datasets of endothelial cells from bone and muscle in healthy human donors. However, since we recognize the importance of the issue, we tried to address it in different and complimentary ways: First, we compared the gene expression of our bEC and mEC with the expression of healthy bEC and mEC from mouse, found in the literature (dataset GSE47067, linked to the paper Nolan DJ, Ginsberg M, Israely E, Palikuqi B et al. Molecular signatures of tissue-specific microvascular endothelial cell heterogeneity in organ maintenance and regeneration. Dev Cell 2013 Jul 29;26(2):204-19). Our analysis of upregulated genes in both bone and muscle endothelium in OA humans compared to healthy mice evidenced no correlation between the two data sets. Indeed, only two genes were upregulated in both human and mouse muscle endothelium and no common upregulated or downregulated gene was found for bone. This striking difference between human and mouse endothelial cells finds support in several studies, highlighting how endothelial cells from many tissues (e.g. lymphnode, brain, liver) behave differently between mouse and humans (e.g. Xiang M et al, Front. Cardiovasc. Med 2020; Song H et al, Sci Rep 2020, Ohnishi et al, Toxicol Pathol, 2007). Our different transcriptomic results were probably due to several reasons: beside species specific differences between human and mouse (e.g. TNFRSF6b, upregulated in bECs, does not have a mouse ortholog),

there were technical differences (different platforms, cell isolation and culture methods ecc..) and disease status. Thus, we were not able to draw meaningful conclusions on the origin of differences found between our bEC and mEC and did not include those results in the revised paper.

We then decided to compare our bECs/mECs expression profiles with those of HUVECs as healthy controls. This analysis aimed at evidencing possible genes/processes equally up or downregulated in both mEC and bEC by OA conditions. We retrieved six transcriptomic profiles of HUVECs from two publicly available datasets on GEO (GSE144715 and GSE108723), which used the same Agilent platform as ours for the gene expression analysis, in order to limit biases during their integration. We then compared gene expression from bECs/mECs with that of HUVECs: only one gene was significantly up-regulated in the bECs vs HUVEC comparison (SELE-E, also up-regulated in bECs vs mECs) and two in the mECs vs HUVEC comparison (RELN and POSTN, also up-regulated in mECs vs bECs). Since no new up- or down-regulated genes were seen in the comparison of bECs/mECs with HUVEC, from this additional finding we can conclude that osteoarthritis does not modulate gene expression in a uniform way on the two different endothelia, when compared with a normal cell line as control.

Since however we could not exclude that pathological conditions differently influence the transcriptomic profile of both endothelial cells, we performed a further comparison among endothelial cells from patients with a different degree of OA severity, calculated as Tonnis score. We divided our bECs and mECs in two groups, according to OA grade, and observed gene expression and angiogenic potential of both mECs and bECs. We are aware that our patients were all end-stage disease patients, thus without striking differences among the groups, however, the comparison between the them could evidence possible influences of progressing pathological conditions on endothelial behavior. The angiogenic potential resulted slightly higher for bECs (but not for mECs) derived from more severe OA patients as compared to less severe OA. On the other hand, a few genes were differently regulated between ECs of the two OA grade groups. To cite a few, CCL2 and LAMC2 were upregulated in grade 3 ECs vs grade 2, whilst CDH13 was downregulated. The fact that these genes were all different from those differently regulated between bECs and mECs, with the exception of an interferon receptor, suggests that the results reported in our study are mainly attributable to the different endothelial origin, although also pathological conditions affected endothelial behavior. We added those new results both in the results section (see page 17 line 5-12: *"To highlight a possible effect of the pathology on the observed differential gene expression of bECs vs mEC, we compared*

ECs deriving from patients with different OA grades. We found that there were around 60 genes significantly up or down regulated comparing ECs from patients with OA grade 2 vs. grade 3. Among the genes upregulated in grade 3 ECs as compared to grade 2 ECs, we found cytokines like CCL2 and matrix proteins such as LAMC-2, whilst other genes such as CADH13 were downregulated. Interestingly, only 3 genes related to IFN γ response were also differentially expressed in mECs vs bECs, suggesting distinct effects of the tissue of origin and of pathological conditions..”, page 21 line 7-12: “To investigate if the OA grade influenced the angiogenic potential, particularly in bEC, we compared the same parameters between grade 2 and grade 3 OA derived bEC and mEC. We found that bEC derived from grade 3 OA patients had a higher branch number and area sprout as compared to bEC derived from grade OA 2 patients, although the difference did not reach statistical significance (Supplementary Figure 4.)” and in the discussion (see page 25 line18-pag 26 line 9: “One limitation of our study is related to the unavailability of healthy subject-matched bone and muscle endothelia as control conditions, thus, to highlight possible effects due to the pathology, we compared ECs from patients with different OA degrees, although being all end-stage disease patients. We found that the genes differentially expressed between mEC and bEC were not shared with those differentially expressed between OA grade 2 and grade 3 ECs. Furthermore, when we compared the transcriptomic profile of OA ECs with that of healthy HUVECs (data not shown), we did not find possible signatures induced by OA in both ECs as compared to a healthy endothelium. Taking these considerations together, we can suppose that the differential expression observed between OA bECs and mECs is mainly attributable to the different origin of endothelial cells, as widely reported in the literature for other healthy tissue types and organisms [2, 4].”)

New supplementary Figure 4 showing results on angiogenic potential of bEC and mEC from different OA grade patients

In Table 1, patient information such as grade of OA and degree of sarcopenia (grip strength and muscle mass) should be provided.

We thank the reviewer for the suggestion. We added information about OA grade, expressed as Tonnis score, for all the patients involved in the study (see revised version of Table 1, pag 10). However, we could not add data on

	sarcopenia, since the relevant parameters were not evaluated during patient's hospital stay in the orthopedic surgery department.
In the legend of Figure 1P and Q, the number of patients is described as n=6, but the dots in the graph are n=5. In supplementary figure 1, the dots on the graph is n=4. Please correct them	We thank the reviewer for the comment, we corrected the figure legends according to the graphs (page 13, lines 4-5 and line 15)
In Figure 2A, please provide a description of the columns below the clustering diagram, such as "bEC/mEC" or "Patient"	We added a description of the data grouping in the figure legend (page 16, lines 3-6): "Data were grouped following the cell type (endothelial cells: red squares, non-endothelial cells: green squares), the tissue of origin (bone: bEC, light purple squares, muscle: mEC, dark purple squares), the patient of origin and the sex of the patient (males: light blue squares, females: pink squares)."
In the legend of Figure 2, labels "B" and "C" are interchanged	Thanks for the correction. We modified the legend, exchanging the descriptions of the two labels (page 16, lines 6-9)
In Table 2, why are only the GSEA results for mECs shown, even though the GO analysis shows results for upregulation of inflammatory pathways in bECs (Figure 1D)? Please provide convincing explanations.	We thank the reviewer for pointing it out. Indeed, we found an increased expression of inflammatory pathways in bEC as compared to mEC, irrespective of patient parameters such as age and CRP. However, when we analyzed a possible correlation between those pathways and CRP values or age in bEC only, we indeed found no significant correlation. On the other hand, in mEC a few processes related to abnormal angiogenesis were positively correlated with CRP values and inflammatory response processes were negatively correlated (those reported in Table 2). This different result can be linked to the apparent lower sensitivity of bEC behavior (both regarding angiogenesis and proliferation) to systemic inflammatory conditions. Looking at Figure 3C, it is possible to see that in bEC there is absence of correlation between angiogenic sprouting and CRP values ($r^2 = 0.25$) whilst in mEC the correlation is almost significant ($r^2 = 0.67$, $p = 0.08$). Even for cell growth, as it can be seen from Figure 1P, the correlation between bEC growth and CRP values ($r^2 = 0.88$) is weaker than that of mEC ($r^2 = 0.98$). Moreover, there is a significant difference ($p < 0.001$) between the two regression lines, with the line interpolating mEC values steeper than the line interpolating bEC results. Taken together, all the results seem to sustain the hypothesis that bEC are less influenced by systemic inflammation values as compared to mEC. We added these considerations in the discussion of the revised version (page 26, line 23-25: "We found that bECs were quite insensitive to CRP values, both in terms of proliferation and angiogenic potential, supporting the absence of correlation between the expression of angiogenic processes in bECs and CRP")
The authors showed that different responses of ECs to TNF α or IFN γ (Figure 4 and	We thank the reviewer for the comment and adjusted the discussion section accordingly. See page 28, lines 6-15:

supplementary figure 4). Interpretations of the different responses of ECs to inflammatory cytokines should be discussed in the discussion section.

“The effects of $TNF\alpha$ and $IFN\gamma$ on endothelial cells angiogenesis have been widely studied in the literature, with the majority of studies describing a pro-angiogenic role of $TNF\alpha$ [38, 39], whilst less consensus exists on the effects of $IFN\gamma$, which have been described as either pro-angiogenic [40] or anti-angiogenic [41], with a detrimental effect on bone endothelium [42]. In our study, $IFN\gamma$ effects were opposite to those of $TNF\alpha$, increasing angiogenesis in mECs while decreasing it in bECs, further evidencing how angiogenic response to cytokine stimulation can be differentially modulated in different endothelia. These results extend those of previous studies on the response of different ECs (HUVECs and HAECs) to cytokines [43] and well fit with the hypothesis of the existence of an organ-specific endothelial inflammation signature, recently shown in the mouse [44].”

Reviewer #3 (Remarks to the Author):

Comment	Answer
In the present study, the authors explored if constitutive differences in organotypic endothelia can cause different functional endothelial responses to inflammatory conditions, which could be at the basis of the alterations in muscle and subchondral bone of osteoarthritis patients. They used an approach combining microfluidic angiogenic assay with genome-wide transcriptome profiling to evidence genomic and functional differences between human endothelial cells, isolated from the subchondral bone and from the skeletal muscle of the same patient. Then, they correlated differences in growth and response to inflammation with osteoarthritis patient characteristics such as systemic inflammation parameters, as CRP, and age. Osteoarthritis induced a pro-inflammatory gene signature in bone derived endothelial cells compared to patient-matched muscle endothelial cells, with an upregulation of processes like cytokines' response and expression of adhesion molecules and TNF-alfa receptors. Angiogenic sprouting of muscle endothelial cells was lower than that of bone endothelial cells in control conditions and after TNF-alfa stimulation. Besides, growth of muscle but not of bone endothelial cells decreased with increasing age and systemic inflammation. The obtained findings indicate that inflammatory conditions in osteoarthritis patients differently affect bone and muscle endothelial cells. Specifically, inflammatory processes possibly increase angiogenesis in subchondral bone and associated systemic low-grade inflammation could impair angiogenesis in muscle. Interestingly, the whole data also provide an explanation for the sarcopenia often observed in osteoarthritis patients with high inflammation levels. Collectively, the presented data are of good quality and experiments are well designed. The manuscript is quite well written and only needs of few corrections and improvements:	We thank the reviewer for the kind appreciation of our work and for its useful comments
Page 9: The authors should specify they enrolled osteoarthritis patients both in the text and in the Table.	As suggested by the reviewer, we specified that the subjects were OA patients, both in the text (page 10, line 2), in Table 1 and in its legend (page 10, line 5)
Figure 1D should be quoted within the manuscript	We thank the reviewer for the comment. We added a reference to figure 1D in the method section (paragraph Angiogenesis assay on chip, page 8 line 21) as follows: "...until an endothelial cell monolayer formed between the chip pillars (Figure 1D)".

The use of microarray as strategy to obtain differential gene expression data should be mentioned in the text of both Results and related Figure legend	As suggested, we added the information on microarray both in the results (page 14, lines 7-8) modifying the sentence as following: “We analyzed transcriptomic profile from 11 patients, 5 males and 6 females, comparing patient-matched RNA from bECs and mECs through whole-genome microarrays.” and in the related figure legend (page 16, line 1) as follows: “Figure 2: Results of microarray transcriptomic analysis. A) Clustering analysis...”
Legend to Figure 2: descriptions of panels B and C are inverted and must be corrected	Thanks for the correction. We modified the legend, exchanging the descriptions of the two labels (page 16, lines 6-9)
Discussion: The authors could improve this section by examining the limitations of their study; for instance, they used microarrays to analyze the whole gene expression instead of NGS strategies (especially single cell transcriptomics), which could provide further insights on the knowledge of “inflamm-aging” and related pathologies if combined with microfluidics	We agree with the reviewer that there are several advanced methods for transcriptome profiling that we could have considered, i.e., NGS based approaches such as single cell sequencing. Single cell technologies revolutionized the transcriptomic analysis providing single cell level information. The application of this technology would have provided a huge novel insight in our work, yet we decided to opt for a transcriptome profiling method that is more robust, reliable, and high throughput, as the microarray. Indeed, although single cell technology would have provided a more comprehensive data set, there are low transcripts genes that could have been undetected using this technology. Also, for the purpose of the manuscript, to characterize the whole endothelial cell populations from different tissues we did not need the single cell data detail. The single cell approach could be implemented in follow up studies where a more in-depth characterization of the system will be needed. In the revised version of the article, we now added these considerations in the discussion section (page 25 lines 12-14) as follows:” Although single cell sequencing would have provided a more comprehensive data set, we opted for microarray analyses since low transcripts genes may have gone undetected using the former technology.”

Reviewer #4 (Remarks to the Author):

Comment	Answer
The authors describes the finding of OA patients (17) samples that shows that inflammatory processes increase angiogenesis in subchondral bone while associated systemic low-grade inflammation impairs angiogenesis in muscle. These results are derived mainly from gene expression, immunohistochemistry and angiogenesis assays. Overall the findings of the study are interesting and the experiments are technically sound. I am not an expert in the research field so won't comment on the impact and significance of the work. I have following technical concerns, which need to address before accepting the manuscript for publication.	We thank the reviewer for the kind appreciation of our work and for its valuable comments
Inflammation is directly linked with the BMI and if the 17 samples are analyzed based on group of low, high and very high BMI, It is advisable that gene expression data also re-analyzed in groups of BMI values. Although the sample size is small here, it might indicate the correlation of BMI and inflammation linked markers.	We thank the reviewer for highlighting the possible correlation between BMI and gene expression data. As suggested, we analyzed our data to look for biological processes that could have been correlated, either positively or negatively, with BMI in both bEC and mEC. Surprisingly, we did not find a correlation between CRP and BMI in our dataset (we added the analysis at page 10, line 9, as follows: “[...] values higher for some obese patients, although CRP and BMI values were not correlated ($r^2 = 0.19$).”), so we analyzed the correlation between bEC and mEC gene expression and BMI as a further independent variable. In bEC we previously find no correlation between angiogenesis or inflammation related processes and CRP values. On the other hand, we found processes related to angiogenesis that were positively correlated with BMI and anti-inflammatory processes (negative regulation of TNF production or production of anti-inflammatory cytokine production) which were negatively correlated with BMI. This last result indeed supports a link between increasing BMI and increasing inflammation. Regarding the results about mEC, we highlighted a few processes related to abnormal angiogenesis and decreased muscle differentiation which were positively correlated with CRP values and inflammatory response processes which were negatively correlated. When we analyzed the correlation with BMI we found no process related to inflammation or angiogenesis which was positively or negatively correlated to BMI. These new results are now reported in a modified version of Table 2. We also modified the text (page 17 lines 12-17) as follows: “We found that in mECs some processes, including regulation of inflammatory cytokine production, were positively correlated with age and CRP, whilst processes associated to angiogenesis were negatively correlated. No correlation between angiogenic or inflammatory processes and CRP or age”

	values has been found in bEC, which instead showed a positive correlation of angiogenic processes and a negative correlation of anti-inflammatory processes with BMI (Table 2)." To further analyze the possible correlation between BMI and EC behavior, we also correlated bEC and mEC growth rate with BMI values, without finding a significant r^2 value for neither bEC nor mEC. We reported the results in a modified version of Supplementary Figure 1, and modified the text accordingly (page 12 lines 10-12): "On the contrary, we found no correlation between mEC and bEC growth with BMI values (Supplementary Figure 1B)."   New Supplementary Figure 1 with the addition of the correlation between growth rate and BMI (B)
Discussion of the manuscript is not written clearly. Authors should put forward the prospective of their study in the field	We thank the reviewer for the comment. In this revised version we completely restructured the discussion section.
Immunofluorescence data quality lack clarity	In the original version of the article, we kept the raw images as they were taken from the microscope, possibly resulting in a sub-optimal visualization. In this revised version we uniformly adjusted the image brightness, to highlight the signal without affecting the comparison between the different conditions
All Figures: scale bar is not visible	We thank the reviewer for pointing it out. We now added the scale bars where they were missing and highlighted them more where they were already present. The legends of the figures were corrected accordingly: Figure 1: page 12, line 4, we added "Scale bar: 250 μm" Supplementary Figure 2: page 19, line 3, we added: "Scale bars: 50 μm" Supplementary Figure 3: page 20, line 3, we added: "Scale bars: 100 μm" Supplementary Figure 5 (now Supplementary Figure 6): page 28, line 2, we added: "Scale bars: 50 μm"
S Figure 5: bar in not there and figure legend is not clear, CRP values should be 0.3 not 0,3, similar typo mistake are identified several places	We added the scale bar in the images, that was indeed missing, and we rephrased the legend (page 27, line 17) as follows: "Immunofluorescence images of IGFBP3 expression in cultured mECs. We compared the expression in a representative patient

with high systemic inflammation value (CRP>0.3) with that in a representative patient with low systemic inflammation value (CRP<0.2). The comparison was performed in control conditions (upper row) or when stimulated with TNF α (lower row). Scale bars 50 μ m."

Regarding the decimal separators, we corrected the numbers through the article where the commas were present, as suggested by the reviewer.

New Supplementary Figure 5 (now Supp Fig 6) with the modifications suggested by the reviewer

Figure 4B panel is not clear in reading

Thanks to the reviewer for the correction. We increased the dimensions of the font in the graphs to make them more readable.

New Figure 4B with the modifications suggested by the reviewer

REVIEWERS' COMMENTS:

Reviewer #2 (Remarks to the Author):

The authors have addressed previous concerns. I think ms is now suitable for publication.

Reviewer #3 (Remarks to the Author):

The authors have addressed all my concerns

Reviewer #4 (Remarks to the Author):

The response to the reviewer comments are justified and overall I found the manuscript much improved particularly the discussion section.
I recommend the revised manuscript for the publication now.